# Antitumor Effect of Morusin via G1 Arrest and Antiglycolysis by AMPK Activation in Hepatocellular Cancer

**DOI:** 10.3390/ijms221910619

**Published:** 2021-09-30

**Authors:** Ah-Reum Cho, Woon-Yi Park, Hyo-Jung Lee, Deok-Yong Sim, Eunji Im, Ji-Eon Park, Chi-Hoon Ahn, Bum-Sang Shim, Sung-Hoon Kim

**Affiliations:** Molecular Cancer Target Herbal Research Laboratory, College of Korean Medicine, Kyung Hee University, Seoul 02447, Korea; kitty389@hanmail.net (A.-R.C.); wy1319@naver.com (W.-Y.P.); hyonice77@naver.com (H.-J.L.); simdy0821@naver.com (D.-Y.S.); ji4137@naver.com (E.I.); wdnk77@naver.com (J.-E.P.); ach2565@naver.com (C.-H.A.)

**Keywords:** Morusin, G1 arrest, glycolysis, AMPK, liver cancer

## Abstract

Though Morusin isolated from the root of *Morus alba* was known to have antioxidant, anti-inflammatory, antiangiogenic, antimigratory, and apoptotic effects, the underlying antitumor effect of Morusin is not fully understood on the glycolysis of liver cancers. Hence, in the current study, the antitumor mechanism of Morusin was explored in Hep3B and Huh7 hepatocellular carcninomas (HCC) in association with glycolysis and G1 arrest. Herein, Morusin significantly reduced the viability and the number of colonies in Hep3B and Huh7 cells. Moreover, Morusin significantly increased G1 arrest, attenuated the expression of cyclin D1, cyclin D3, cyclin E, cyclin-dependent kinase 2 (CDK2), cyclin-dependent kinase 4 (CDK4), and cyclin-dependent kinase 6 (CDK6) and upregulated p21 and p27 in Hep3B and Huh7 cells. Interestingly, Morusin significantly activated phosphorylation of the adenosine 5′-monophosphate (AMP)-activated protein kinase (AMPK)/acetyl-CoA carboxylase (ACC) but attenuated the expression of the p-mammalian target of protein kinase B (AKT), rapamycin (mTOR), c-Myc, hexokinase 2(HK2), pyruvate kinases type M2 (PKM2), and lactate dehydrogenase (LDH) in Hep3B and Huh7 cells. Consistently, Morusin suppressed lactate, glucose, and adenosine triphosphate (ATP) in Hep3B and Huh7 cells. Conversely, the AMPK inhibitor compound C reduced the ability of Morusin to activate AMPK and attenuate the expression of p-mTOR, HK2, PKM2, and LDH-A and suppressed G1 arrest induced by Morusin in Hep3B cells. Overall, these findings suggest that Morusin exerts an antitumor effect in HCCs via AMPK mediated G1 arrest and antiglycolysis as a potent dietary anticancer candidate.

## 1. Introduction

Hepatocellular cancers (HCCs) are known the most common type of primary liver cancers worldwide. Though chemotherapy, radiotherapy, immunotherapy, and surgery have been used for treatment of HCCs for years [1], antiglycolytic agents including 3-bromopyruvate are of interest for the treatment of HCCs in combination with classical anticancer agents such as sorafenib or alone [2,3].

Emerging evidence reveals that metabolic intermediates of glycolysis, known as “the process of conversion of glucose into pyruvate followed by lactate production”, facilitate cancer proliferation and progression [4,5], which was defined as the Warburg effect [6,7]. Among several cancers, liver cancer progression and resistance are more associated with metabolic glycolysis [8], since aerobic glycolysis induces an acidic environment to aid proliferation, angiogenesis, invasion, metastasis, and immune evasion in HCC [9].

Accumulating evidence indicates that phosphofructokinase 1 (PFK1), hexokinase 2 (HK2), and pyruvate kinases type M2 (PKM2) are critically involved in aerobic glycolysis in HCC in association with other signaling networks such as the PI3K/Akt/mTOR pathway, HIF-1α, c-Myc, AMPK, and noncoding RNAs [10].

Recently, natural compounds are of interest as potent adjuvants to cancer therapy with lower toxicity and fewer side effects [11]. Indeed, several natural compounds such as muscone [12], oleanolic acid [13], isobavachalcone [14], baicalein [15], astrakurkurone [16], and others were found to have anticancer potential in liver cancers [17]. Likewise, though Morusin derived from the roots of *Morus alba* was found to have anticancer effects in several cancers, the underlying antitumor mechanisms still remain unclear. Hence, in the present study, the antitumor mechanism of Morusin was examined in Hep 3B and Huh7 HCCs in association with cell cycle arrest and glycolysis related signaling.

## 2. Results

### 2.1. Morusin Increases Cytotoxicity and Induces G1 Arrest in Huh7 and Hep3B Cells

To check the cytotoxicity of Morusin (Figure 1A), MTT, CCK-8, and colony formation assays were carried out in Huh7 and Hep3B cells. As shown in Figure 1B,C, Morusin significantly reduced the viability of Huh7 and Hep3B cells in a concentration and time dependent manner by MTT and CCK-8 assay. Consistently, Morusin significantly abrogated the number of colonies in Huh7 and Hep3B cells by colony formation assay as a long-term cytotoxicity assay (Figure 1D).

### 2.2. Morusin Increases G1 Arrest and Modulates G1 Related Proteins in Huh7 and Hep3B Cells

To test whether the cytotoxicity of Morusin was due to cell cycle arrest, cell cycle analysis was performed in Huh7 and Hep3B cells. As shown in Figure 2A,B, Morusin significantly increased G1 portion compared to the untreated control in Huh7 and Hep3B cells (Figure 2A,B). Hence, the effect of Morusin was evaluated in Huh7 and Hep3B cells by Western blotting. Here Morusin significantly abrogated the expression of Cyclin D1, CyclinD3, Cyclin E, CDK2, CDK4, and CDK6 and upregulated p21 and p27 in Huh7 and Hep3B cells (Figure 2C,D).

### 2.3. Morusin Reduces Glycolysis Related Proteins in Huh7 and Hep3B Cells

It is well documented that cancer cells are more active in glycolysis to obtain more ATP for metabolic activity than normal cells, which is named Warburg [18]. Thus, to check the effect of Morusin on glycolysis related proteins, Western blotting was performed in in Huh7 and Hep3B cells. Herein, Morusin significantly activated phosphorylation of AMPK/ACC but attenuated the expression of p-mTOR, p-AKT, cMyc, HK2, PKM2, and LDH-A in Huh7 and Hep3B cells (Figure 3A,B). Consistently, Morusin suppressed lactate, glucose, and ATP in Hep3B and Huh7 cells (Figure 3C).

### 2.4. The Pivotal Role of AMPK in the Antitumor Effect of Morusin in Hep3B Cells

To confirm the role of AMPK in the antitumor effect of Morusin, an inhibitor study was conducted with AMPK inhibitor compound C in Hep3B cells. The cytotoxicity by Morusin was significantly rescued by compound C in Hep3B cells (Figure 4A).

Herein, compound C reduced the ability of Morusin to activate AMPK and attenuate the expression of p-mTOR, p-ACC, HK2, PKM2, and LDH-A in Hep3B cells (Figure 4B). Consistently, compound C suppressed G1 arrest induced by Morusin in Hep3B cells (Figure 4C). Conversely, suppressed glucose, lactate, and ATP by Morusin was significantly reversed by compound C in Hep3B cells (Figure 4D).

## 3. Discussion

Generally, three common phonotype enzymes, hexokinase 2 (HK2), phosphofructokinase 1 (PFK1), and pyruvate kinases type M2 (PKM2) were known in the glycolytic process of HCCs. Moreover, AMPK, PI3K/Akt pathway, HIF-1α, and c-Myc have emerged as aerobic glycolysis related proteins in hepatocellular carcinoma [10].

In the current study, the antitumor mechanism of Morusin was investigated in Huh7 and Hep3B cells in association with glycolysis and G1 arrest. Usually MTT and CCK-8 assays have been used for short term cytotoxicity [19,20], while colony formation has been applied for long term cytotoxicity [21,22]. Here, Morusin significantly showed cytotoxicity by MTT and CCK-8 assays and abrogated the number of colonies in Huh7 and Hep3B cells, implying the antitumor potency of Morusin. Likewise, Morusin induced apoptosis and inhibited angiogenesis in hepatocellular carcinoma cells by inhibiting the IL6/STAT3 signaling pathway [23]. Moreover, previous studies demonstrated that Morusin has an antitumor effect in several cancers such as lung [24], breast [25], renal [26], and pancreatic cancers [27].

It is well documented that cell proliferation depends on four distinct phases of the cell cycle including G0/G1, S, G2, and M, which is usually regulated by several cyclin-dependent kinases (CDKs) [28]. Cell cycle arrest is known as a stopping point in the cell cycle through the G_1_ phase, S phase (synthesis), G_2_ phase (interphase), and M phase (mitosis and cytokinesis) for DNA duplication or cell division [29,30]. The expression of p21^Cip1/WAF1^ maintains pRb and induces G1 arrest [31]. Moreover, mitogenic signals induce cyclin D1 expression for binding to CDK4/or CDK6 in the G1 phase of the cell cycle, while cyclin E and CDK2 are activated in the late G_1_ phase [32,33]. Here, Morusin significantly increased G1 arrest in Huh7 and Hep3B cells. Consistently, Morusin reduced the expression of cyclin D1, cyclin D3, cyclin E, CDK2, CDK4, and CDK6 and upregulated p21 and p27 in Huh7 and Hep3B cells, indicating cytotoxicity is induced partially due to G1 arrest by Morusin.

Accumulating evidence reveals that cancer cells rapidly consume glucose and convert it into lactate for uncontrolled replication and proliferation [30]. Among the metabolic intermediates of glycolysis, fructose-1, 6 bisphosphate enhances the antiapoptotic effect in cancer cells by reducing the release of cytochrome C [34], while pyruvate facilitates chemoresistance by overexpression of p-glycoprotein, leading to the export and import of lactate as the product of pyruvate oxidation [35]. Moreover, it is well documented that AMPK as a conserved and ubiquitously expressed energy sensor reduces the phosphorylation of mTOR and c-Myc [36], leading to inhibition of glycolysis and cell proliferation [36,37]. Recent studies demonstrate that AKT promotes cancer progression and the Warburg effect, since it facilitates glycolysis in cancer cells [38]. The accumulation of lactate in cancer cells promotes proliferation and growth, implying that inhibition of AKT by Morusin suppresses the growth and survival of liver cancer cells via the reduction in the Warburg effect. Herein, Morusin significantly activated phosphorylation of AMPK/ACC but attenuated the expression of p-AKT, p-mTOR, c-Myc, HK2, PKM2, and LDH-A and suppressed lactate, glucose, and ATP in Huh7 and Hep3B cells, strongly demonstrating the antiglycolytic effect of Morusin in HCCs. Notably, compound K suppressed glycolysis in hepatocellular cells [38], and Lapachol inhibited glycolysis in melanoma cancer cells [39]. Moreover, Morusin induced autophagy in lung cancer cells by activating AMPK [40].

AMPK can be a target to control glycolysis in several cancers, since AMPK is activated to inhibit biosynthetic pathways that consume ATP and enhance ATP-generating pathways involving glucose transport and glycolysis [41]. Thus, AMPK activator Metformin has been used in several cancers related to glycolysis [42]. To confirm the pivotal role of AMPK in the Morusin induced antitumor effect, AMPK inhibitor compound C was used in Morusin-treated Hep3B cells. As expected, compound C suppressed the capability of Morusin to activate AMPK, inactivate ACC, attenuate the expression of p-mTOR, HK2, PKM2, and LDH-A, and induce G1 arrest in Hep3B cells, indicating the important role of AMPK in the Morusin-exerted antitumor effect. Conversely, compound C rescued the suppression of lactate, glucose, and ATP by Morusin in Hep3B and Huh7 cells. Moreover, Gao et al. [17] reported that Morusin exerted an apoptotic effect in HepG2 and Hep3B cells and the HepG2 xenograft model via inhibition of STAT3 phosphorylation, implying the therapeutic potential of Morusin.

In summary, Morusin significantly abrogated the viability and the number of colonies, increased G1 arrest, attenuated the expression of cyclin D1, cyclin D3, cyclin E, CDK2, CDK4, and CDK6 and upregulated p21 and p27 in Huh7 and Hep3B cells. Moreover, Morusin significantly activated phosphorylation of AMPK/ACC and attenuated the expression of p-AKT, p-mTOR, c-Myc, HK2, PKM2, and LDH-A in Huh7 and Hep3B cells. Conversely, the AMPK inhibitor compound C reduced the ability of Morusin to activate AMPK, inactivate ACC, attenuate the expression of p-mTOR, HK2, PKM2, and LDH-A and induce G1 arrest in Morusin-treated Hep3B cells (Figure 5). Taken together, these findings provide evidence that Morusin exhibits an antitumor effect in HCCs via AMPK mediated G1 arrest and antiglycolysis as a potent dietary anticancer candidate.

## 4. Materials and Methods

### 4.1. Morusin

Morusin (CAS # 62596-29-6) with a purity of over 99% was purchased from Sigma–Aldrich (Sigma–Aldrich, St Louis, MO, USA), diluted in 0.1% DMSO and stored for the next experiments.

### 4.2. Cell Culture

Human hepatocellular cancer cell lines including Hep3B (*ATCC* HB-8064^)^ and Huh7 (PTA-4583) cells were bought from American Type Culture Collection (ATCC). The cells were maintained in DMEM and RPMI1640 with 10% FBS and 1% antibiotic (Welgene, South Korea).

### 4.3. Cytotoxicity Assay (MTT and CCK-8 Assay)

The cytotoxicity of Morusin was assessed by using a 3-(4,5-dimethylthiazol-2-yl)-2,5-diphenyltetrazolium bromide (MTT) assay. Briefly, Huh7 and Hep3B cells (1 × 10^4^ cells/well) were exposed to various concentrations (0, 2.5, 5, 10, 20, 40 μM) of Morusin for 24 h, incubated with MTT (1 mg/mL) (Sigma Chemical, St Louis, MO, USA) for 2 h and then treated with MTT lysis solution overnight. Then, cell viability was calculated as a percentage of viable cells in the Morusin-treated group versus the untreated control by detecting optical density by using a microplate reader (Molecular Devices Co., San Jose, CA, USA) at 570 nm. Also, HCCs were exposed to 10 µL CCK-8 reagent and were incubated at 37 °C for 2 h. The optical density (OD) values were determined using a microplate reader at 450 nm.

### 4.4. Colony Formation Assay

Huh7 and Hep3B cells were seeded onto 6-well plates at a density of 1000 cells per well for a week. Then the number of colonies stained with crystal violet were counted.

### 4.5. Cell Cycle Analysis

Based on Gao et al.’s paper [13], Huh7 and Hep3B cells (1 × 10^6^ cells/well) were exposed to Morusin (0, 2.5, or 5 μM) for 24 h and fixed in 75% ethanol at −20 °C. The cells were incubated with RNase A (10 mg/mL) for 1 h at 37 °C and stained with propidium iodide (50 μg/mL) for 30 min in the dark. The stained cells were analyzed for DNA content by FACSCalibur (Becton Dickinson, Franklin Lakes, NJ, USA).

### 4.6. Western Blotting

Based on Kim et al.’s paper [43], Huh7 and Hep3B cells (1 × 10^6^ cells/mL) exposed to various concentrations of Morusin for 24 h were lyzed in lysis buffer on ice. The supernatants obtained by centrifugation were quantified for protein concentration and were transferred to a Hybond ECL transfer membrane for detection with antibodies of CyclinD1, CyclinD3, CyclinE, p21, CDK2, CDK4, p-AKT, AKT, p-mTOR, mTOR, p-AMPK, p-ACC, ACC, HK2, PKM2, LDH-A, and c-Myc (Cell signaling Technology, Beverly, MA, USA) and CDK6, LDH, and p27 (Santa Cruz Biotechnologies, Santa Cruz, CA), and β-actin (Sigma–Aldrich, St. Louis, MO, USA).

### 4.7. Lactate, Glucose and ATP Assay

Cells (1 × 10^6^ cells/ml) were treated with Morusin for 24 h and then cell culture medium was collected. Lactate (K-607), glucose (K-606), and ATP (K-354) in media were measured using a colorimetric assay kit (BioVision Inc., Milpitas, CA, USA) according to manufacturer instructions.

### 4.8. Statistical Analysis

For statistical analysis of the data, all data analyzed by using GraphPad Prism software (Version 5.0, California, USA) were expressed as means ± standard deviation (SD). A Student’s *t*-test was used for comparison of the two groups and the statistically significant difference was determined as *p* values of <0.05 between the control and Morusin-treated groups.

## Figures and Tables

**Figure 1 ijms-22-10619-f001:**
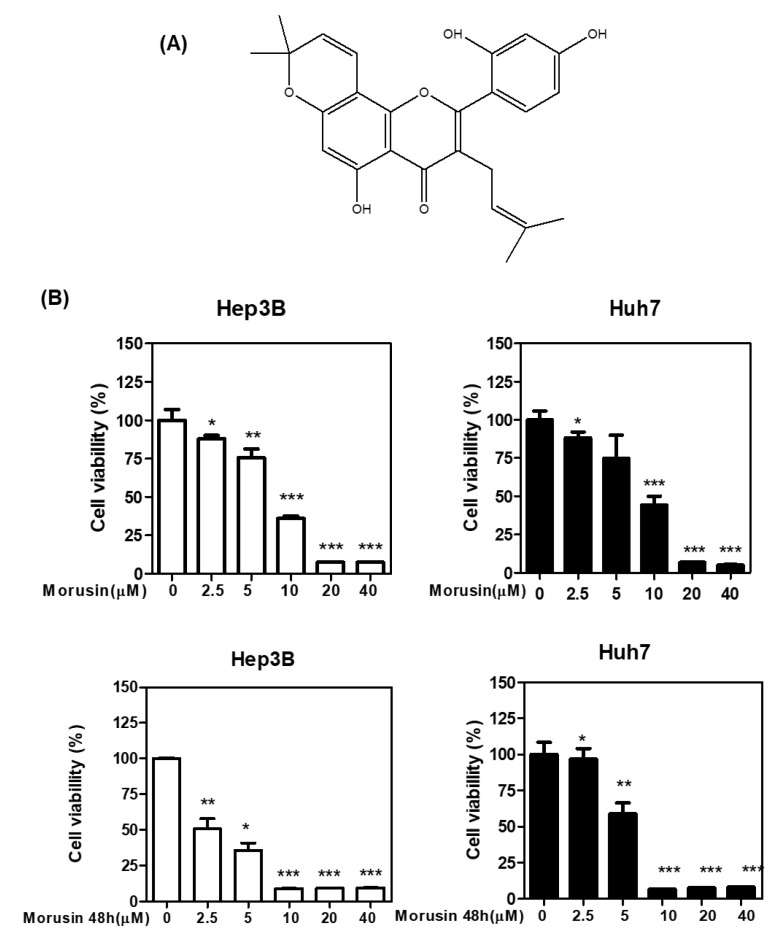
Chemical structure of Morusin and its effect on cytotoxicity in HCCs. (**A**) Chemical structure of Morusin (**B**) Effect of Morusin on the viability of Huh7 and Hep3B cells by MTT assay. Huh7 and Hep3B cells were seeded into 96-well microplates and treated with various concentrations (2.5, 5, 10, 20, and 40 µM) of Morusin for 24 h and 48 h. Cell viability was measured by MTT assay. * *p* < 0.05, **, *p* < 0.01, ***, *p* < 0.001 vs. untreated control. (**C**) Effect of Morusin on Huh7 and Hep3B cells. Cells were cultured with Morusin (0–40 μM) for 24 h and 48 h and then measured by CCK-8 assay (**D**) Effect of Morusin on the number of colonies in Hep3B cells by colony formation assay. Huh7 and Hep3B cells were seeded onto 6-well plates for a week. Then the colonies stained with crystal violet were counted. Data represent means ± SD. *, *p* < 0.05, **, *p* < 0.01, ***, *p* < 0.001 vs. untreated control (*n* = 3).

**Figure 2 ijms-22-10619-f002:**
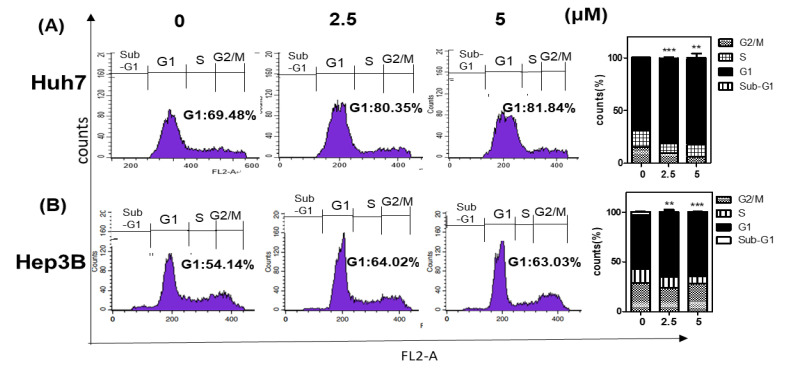
Effect of Morusin on G1 arrest and its related proteins in Huh7 and Hep3B cells. (**A**) Effect of Morusin on G1 arrest in Hep3B cells. *, *p* < 0.05, ***, *p* < 0.001 vs. untreated control. (**B**) Effect of Morusin on G1 arrest in Huh7 cells. Huh7 and Hep3B cells exposed to Morusin (2.5, 5 μM) for 24 h were incubated with RNase A (10 mg/mL) for 1 h at 37 °C and stained with propidium iodide (50 μg/mL) for 30 min at room temperature in the dark. The stained cells were analyzed for DNA content by FACSCalibur. **, *p* <0.01, ***, *p* <0.001 vs. untreated control (**C**) Effect of Morusin on G1 related proteins in Hep3B cells *, *p* < 0.05, **, *p* < 0.01, ***, *p* < 0.001 vs. untreated control. (**D**) Effect of Morusin on G1 related proteins in Huh7 cells. Huh7 and Hep3B cells were treated with Morusin (2.5, 5 μM) for 24 h. Cell extracts were prepared and subjected to Western blotting with Cyclin D1, Cyclin D3, Cyclin E, p21, p27, CDK2, CDK4, and CDK6 antibodies. β-actin was used as an internal control. Data represent means ± SD. *, *p* < 0.05, **, *p* < 0.01, ***, *p* < 0.001 vs. untreated control (*n* = 3).

**Figure 3 ijms-22-10619-f003:**
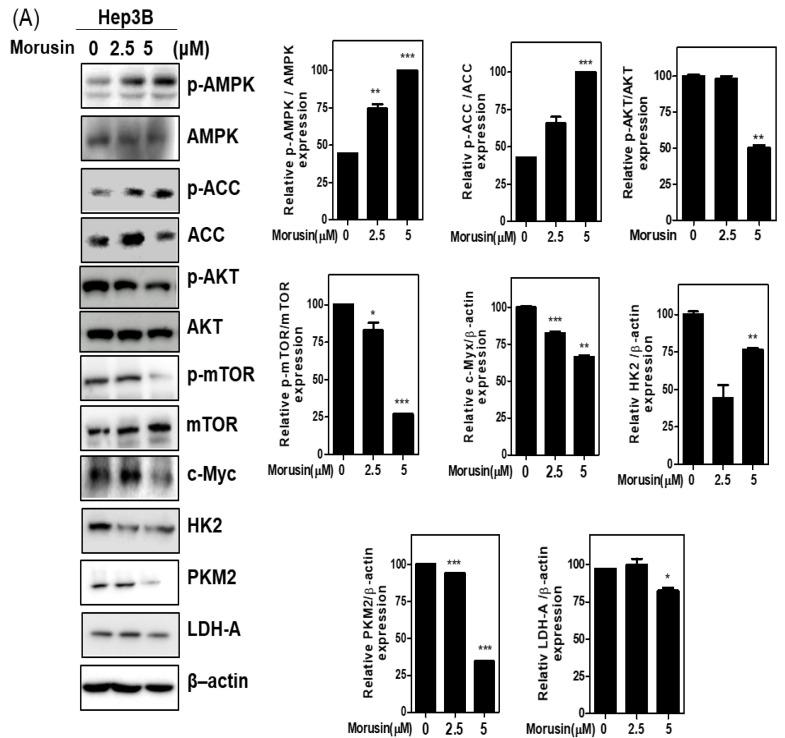
Effect of Morusin on glycolysis related proteins in Hep3B (**A**) and Huh7 (**B**) cells. Huh7 and Hep3B cells were treated with Morusin (2.5, 5 μM) for 24 h. Cell extracts were prepared and subjected to Western blotting with the antibodies of p-AMPK, AMPK, p-AKT, AKT ACC p-ACC, ACC, p-mTOR, mTOR, c-Myc, HK2, PKM2, LDH-A, and β-actin. (**C**) Glucose, lactate, and ATP in conditioned medium were measured using colorimetric assay kit. Data represent means ± SD. *, *p* < 0.05, **, *p* < 0.01, ***, *p* < 0.001 vs. untreated control (*n* = 3).

**Figure 4 ijms-22-10619-f004:**
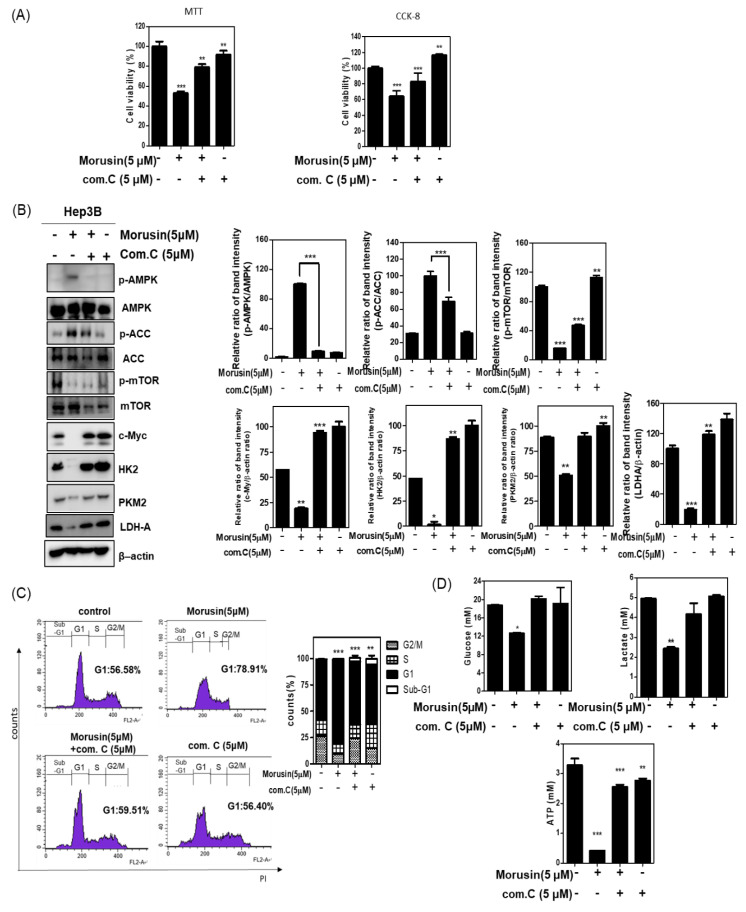
Effect of compound C on glycolysis related proteins and G1 arrest in Morusin treated Hep3B cells. (**A**) Hep3B cells were treated with Moursin in the absence and presence of compound C. Cell viability was evaluated using MTT and CCK-8 assays. (**B**) Effect of AMPK inhibitor compound C on glycolysis related proteins in Morusin treated Hep3B cells. Hep3B cells were treated with Morusin (5 μM) in the presence or absence of compound C (5 μM) for 24 h. Cell extracts were prepared and subjected to Western blotting with the antibodies of p-mTOR, mTOR, p-ACC, ACC p-AMPK, AMPK, c-Myc, HK2, PKM2, and β-actin. Data represent means ± SD. *, *p* < 0.05, **, *p* < 0.01, ***, *p* < 0.001 vs. untreated control. (**C**) Effect of compound C on G1 arrest in Morusin treated Hep3B cells. Hep3B cells exposed to Morusin (5 μM) in the presence or absence of compound C for 24 h were subjected to cell cycle analysis by staining with propidium iodide (50 μg/mL) (*n* = 3). (**D**) Glucose, lactate, and ATP in conditioned medium were measured by using colorimetric assay kit. Data represent means ± SD. *, *p* < 0.05, **, *p* < 0.01, ***, *p* < 0.001 vs. untreated control (*n* = 3).

**Figure 5 ijms-22-10619-f005:**
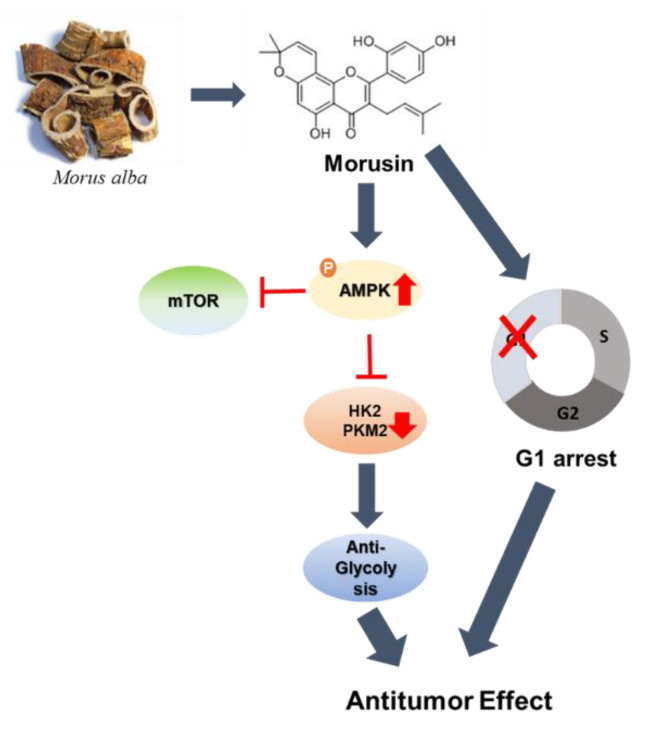
Schematic diagram of the antitumor mechanism of Morusin via G1 arrest and antiglycolysis by AMPK activation.

## Data Availability

All the data and materials supporting the conclusions are included in the main paper.

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
