# Peer review of "Antitumor Effect of Morusin via G1 Arrest and Antiglycolysis by AMPK Activation in Hepatocellular Cancer"

_ijms, 2021, doi:10.3390/ijms221910619_

Round 1

Reviewer 1 Report

  1. Please verify the therapeutic efficacy and potential mechanism of morusin in HCC in vivo.
  2. Please demonstrate change of total protein level of mTOR and ACC in Fig 3A-B.

Author Response

  1. Please verify the therapeutic efficacy and potential mechanism of morusin in HCC in vivo.

(Response) Thanks. But based on antitumor efficacy of morusin in vitro we will perform further animal study with more amount of morusin isolated from Morus alba in the near future.
2.          Please demonstrate change of total protein level of mTOR and ACC in Fig 3A-B.

(Response) Thanks. Total protein blots of mTOR and ACC were added in Fig 3A-B.

Reviewer 2 Report

The MS assesses the effect of the morusin extracted from Morus alba on HCC cells, and especially onthe glycolitic pathways in this cancer. Although the idea may be interesting, some important points are missing:

  1. There are data about the effects of Morusin in HCC and other cancers with signalling pathways well described (https://www.ncbi.nlm.nih.gov/pmc/articles/PMC5481341/). They should be mentioned and discussed.
  2. The origin and purity of Morusin should be mentioned.
  3. Parts of the paragraph from line 59 are missing.
  4. Number of experiments (n) is not reported in MTT assays , flow cytometry or western blot.
  5. The legend in panels from Figure 2 representing cell cycle distribution (M1-M4) should match that from graph representing the averages (G1-G2/M). 
  6. From Fig. 2 it is not clear that the number of cells in G1 are significantly different  in those three conditions presented. Neither the plots, nor the column bar (that do not have Standard deviations) so not support this statment. Proper significance test should be applied and presented in the graphs.The experiments performed are not related to AMPK phosphorylation, therefore the assumption that morusin activates phosphorylation is not sustained.
  7. Compound C should be described and the concentration used for AMPK inhibition should be mentioned.

Author Response

The MS assesses the effect of the morusin extracted from Morus alba on HCC cells, and especially on the glycolytic pathways in this cancer. Although the idea may be interesting, some important points are missing:

  1. There are data about the effects of Morusin in HCC and other cancers with signalling pathways well described (https://www.ncbi.nlm.nih.gov/pmc/articles/PMC5481341/). They should be mentioned and discussed.

(Response) Thanks for your critical comments. The effect of morusin in several cancers was added in Introduction and Discussion.

2.The origin and purity of Morusin should be mentioned.

(Response) Thanks. Added.

3.Parts of the paragraph from line 59 are missing.

(Response) Thanks for your kind comments. Corrected.

4.Number of experiments (n) is not reported in MTT assays, flow cytometry or western blot.

(Response) Thanks. Added.

5.The legend in panels from Figure 2 representing cell cycle distribution (M1-M4) should match that from graph representing the averages (G1-G2/M). 

(Response) Thanks for your kind comments. Corrected.

6.From Fig. 2 it is not clear that the number of cells in G1 are significantly different  in those three conditions presented. Neither the plots, nor the column bar (that do not have Standard deviations) so not support this statement. Proper significance test should be applied and presented in the graphs. The experiments performed are not related to AMPK phosphorylation, therefore the assumption that morusin activates phosphorylation is not sustained

(Response) Thank you, honorable reviewer. However, as shown in Figure 3, we confirm Morusin activates phosphorylation of AMPK. Also, inhibitor study revealed the pivotal role of AMPK in morusin induced apoptosis in Hep3B cells (Figure 4B).

7.Compound C should be described and the concentration used for AMPK inhibition should be mentioned.

(Response) Thanks. Added in Figure4 legend.

Reviewer 3 Report

The manuscript by Cho et al. is aimed at demonstrating the anti-tumor effects of morusin in hepatocellular cancer. However, major points need to be addressed in order to improve the quality of the manuscript:

Introduction: please add literature data about the anti-tumor effects of morusin in various malignancies, clarifying the aim of the present work.

Fig 1: please add MTT assay results after 48h of treatment in order to get both dose- and time-dependent cytotoxic effects. Moreover, please conduct additional assays to confirm these data (Trypan Blue or LDH assay or CCK8 assay).

Fig 2: what is the effect of morusin on Cyclin D3, CDK6 and p27? What is the effect of morusin on apoptosis (for instance, Ann V/PI assay, mitochondrial membrane loss, modulation of Bax/Bcl-2, cytochrome c and caspase expression)?

Fig 3: what is the effect of morusin on Akt phosphorylation? Please measure glucose consumption, lactate production and ATP synthesis after morusin treatment to confirm its inhibitory effect on glycolysis. Please show Western blot data about ACC and mTOR total protein expression and quantify protein levels as phosphorylated protein/total protein (do the same with AMPK). Which isoform of LDH has been analyzed? Is it LDH-A?

Fig 4: what is the effect of compound C pretreatment on morusin-mediated cytotoxicity and, possibly, apoptosis induction? Can it affect glucose consumption, lactate production and ATP synthesis? Can it affect mTOR phosphorylation or ACC and LDH expression? Total AMPK levels missing.

Discussion: What is the common metabolic phenotype of hepatocellular carcinoma? Are the pathways examinated in this article commonly mutated in this tumor? Can morusin induce cell cycle arrest or apoptosis in other malignancies? Can it induce AMPK activation or suppress glycolysis in other tumors? Is the AMPK-mediated suppression of glycolysis a commonly employed anti-tumor strategy and has Compound C been used in cancer treatment? Please add this info in the Discussion section. Please add literature data demonstrating the glycolysis-suppressing effects of other natural compounds in tumors, supporting the use of phytochemicals in cancer prevention, particularly in hepatocellular carcinoma.

Author Response

The manuscript by Cho et al. is aimed at demonstrating the anti-tumor effects of morusin in hepatocellular cancer. However, major points need to be addressed in order to improve the quality of the manuscript:

Introduction: please add literature data about the anti-tumor effects of morusin in various malignancies, clarifying the aim of the present work.

Fig 1: please add MTT assay results after 48h of treatment in order to get both dose- and time-dependent cytotoxic effects. Moreover, please conduct additional assays to confirm these data (Trypan Blue or LDH assay or CCK8 assay).

(Response) Thanks. Further study data were added.

Fig 2: what is the effect of morusin on Cyclin D3, CDK6 and p27? What is the effect of morusin on apoptosis (for instance, Ann V/PI assay, mitochondrial membrane loss, modulation of Bax/Bcl-2, cytochrome c and caspase expression)?

(Response) Thanks. It is well documented that cell proliferation depends on four distinct phases of the cell cycle including G0/G1, S, G2, and M, which is usually regulated by several cyclin-dependent kinases (CDKs). Here, Moursin induced G1-phase arrest, which was confirmed by regulating the protein expression of Cyclin D3, CDK6 and p27 as G1-phase-related proteins (Figure 2C, D).

Fig 3: what is the effect of morusin on Akt phosphorylation? Please measure glucose consumption, lactate production and ATP synthesis after morusin treatment to confirm its inhibitory effect on glycolysis. Please show Western blot data about ACC and mTOR total protein expression and quantify protein levels as phosphorylated protein/total protein (do the same with AMPK). Which isoform of LDH has been analyzed? Is it LDH-A?

(Response) Thanks. Recent studies demonstrate that AKT promotes cancer progression and Warburg effect, since it facilitates glycolysis in cancer cells (Elstrom, R. L., Bauer, D. E., Buzzai, M., Karnauskas, R., Harris, M. H.,Plas, D. R., … Thompson, C. B. (2004). Akt stimulates aerobic glycolysis in cancer cells. Cancer Research, 64(11), 3892–3899. Robey, R. B., & Hay, N. (2009). Is Akt the “Warburg kinase”? Akt-energy metabolism interactions and oncogenesis. Seminars in Cancer Biology, 19(1), 25–31.) The accumulation of lactate in cancer cells promotes proliferation and growth, implying that inhibition of AKT by Morusin suppresses the growth and survival of liver cancer cells via reduction of Warburg effect. Also, further experiment data on lactate production, total mTOR/ACC and LHA-A (Cell signaling Cat:#35820) were added in Figure 3B, C.

Fig 4: what is the effect of compound C pretreatment on morusin-mediated cytotoxicity and, possibly, apoptosis induction? Can it affect glucose consumption, lactate production and ATP synthesis? Can it affect mTOR phosphorylation or ACC and LDH expression? Total AMPK levels missing.

(Response) Thanks. Morusin significantly activated phosphorylation of AMPK/ACC in Huh7 and Hep3B cells, strongly demonstrating anti-glycolytic effect of Morusin in HCCs. It is well documented that AMPK as a conserved and ubiquitously expressed energy sensor reduces the phosphorylation of mTOR and c-Myc, leading to inhibition of glycolysis and cell proliferation. To confirm the pivotal role of AMPK in Morusin induced antitumor effect, AMPK inhibitor compound C was used in Morusin treated Hep3B cells. As expected, compound C suppressed the capability of Morusin to activate AMPK and attenuate the expression of HK2 and PKM2 and suppressed G1 arrest induced by Morusin in Hep3B cells, indicating the important role of AMPK in Morusin exerted antitumor effect. Total AMPK blots were added in Figure 3.

Discussion: What is the common metabolic phenotype of hepatocellular carcinoma?

(Response) Thanks. Aerobic glycolysis was firstly found in HCC as a hallmark of liver cancer progression. Generally three common phonotype enzymes were known as hexokinase 2 (HK2), phosphofructokinase 1 (PFK1), and pyruvate kinases type M2 (PKM2) in the glycolytic process of HCCs. Also, AMPK, PI3K/Akt pathway, HIF-1α, c-Myc have emerged as aerobic glycolysis related proteins in hepatocellular carcinoma. Journal of Experimental & Clinical Cancer Research volume 39, 126 (2020))

Are the pathways examinated in this article commonly mutated in this tumor?

(Response) Thanks for your critical comments, which can be a new project in the future, since we can expect AMPK isomers, fatty liver related novel genes including NEDD or c-Myc can be involved in glycolysis of HCCs. Also, a lot of somatic mutations were reported in most solid tumors (Precision medicine for hepatocellular carcinoma: driver mutations and targeted therapy. Oncotarget. 2017 Aug 15; 8(33): 55715–55730.)

Can morusin induce cell cycle arrest or apoptosis in other malignancies? Can it induce AMPK activation or suppress glycolysis in other tumors? Is the AMPK-mediated suppression of glycolysis a commonly employed anti-tumor strategy and has Compound C been used in cancer treatment?

(Response) Thanks. We agree with you that morusin may induce cell cycle arrest or apoptosis in other cancers, which should be confirmed in the future. As you know, AMPK can be a target to control glycolysis of several cancers, since AMPK is activated to inhibit biosynthetic pathways that consume ATP and enhance ATP-generating pathways involving glucose transport, glycolysis (J Surg Oncol,2012 Nov;106(6):680-8). Indeed, AMPK activator, Metformin has been used in cancers related to glycolysis (Clin Cancer Res. 2011 Jun 15;17(12):3993-4005).

Please add this info in the Discussion section. Please add literature data demonstrating the glycolysis-suppressing effects of other natural compounds in tumors, supporting the use of phytochemicals in cancer prevention, particularly in hepatocellular carcinoma.

(Response) Thanks for your comments. Added in Discussion.

Round 2

Reviewer 1 Report

Please verify the therapeutic efficacy and potential mechanism of morusin in HCC in vivo.

Author Response

Please verify the therapeutic efficacy and potential mechanism of morusin in HCC in vivo.

(Response) Thanks for your critical comments. Gao et al reported that Morusin exerted growth inhibition effects on human HCC cells (HepG2 and Hep3B) in vitro and HepG2 xenograft model. (Morusin shows potent antitumor activity for human hepatocellular carcinoma in vitro and in vivo through apoptosis induction and angiogenesis inhibition. Drug Des Devel Ther. 2017 Jun 16;11:1789-1802.). Herein, Morusin suppressed constitutive as well as IL-6-induced STAT3 phosphorylation in HCC cells and tumor tissues, implying therapeutic potential of Morusin, which was added in Discussion.

Reviewer 2 Report

The problems I raised were correctly addressed. i recommend publication.

Author Response

Thanks a lot for your positive comments

Reviewer 3 Report

The manuscript by Cho et al. has not been significantly improved, and several major points still need to be addressed:

  • Fig. 2: Western Blot data about Cyclin D3, CDK6 and p27 expression have not been added.
  • Fig. 3: glucose consumption and ATP synthesis after morusin treatment need to be measured to confirm its inhibitory effect on glycolysis. The effect of morusin on Akt phosphorylation should be examined by Western Blot.
  • Fig 4: please assess the effect of compound C pretreatment on morusin-mediated cytotoxicity by MTT and CCK-8 assay. Can it affect glucose consumption, lactate production and ATP synthesis? Can it affect mTOR phosphorylation or ACC and LDH expression (Western blot analysis)? Total AMPK levels still missing.
  • Discussion: please add your responses to my previous questions in the discussion section to improve it and support your data.

Minor revisions:

Fig. 1C: X axis is not correct, pease set a thick interval and adjust the graph accordingly.

Author Response

The manuscript by Cho et al. has not been significantly improved, and several major points still need to be addressed:

Fig. 2: Western Blot data about Cyclin D3, CDK6 and p27 expression have not been added.

(Response) Thanks. Added in Figure2.

Fig. 3: glucose consumption and ATP synthesis after morusin treatment need to be measured to confirm its inhibitory effect on glycolysis. The effect of morusin on Akt phosphorylation should be examined by Western Blot.

(Response) Thanks. Added in Figure 3.

Fig 4: please assess the effect of compound C pretreatment on morusin-mediated cytotoxicity by MTT and CCK-8 assay. Can it affect glucose consumption, lactate production and ATP synthesis? Can it affect mTOR phosphorylation or ACC and LDH expression (Western blot analysis)? Total AMPK levels still missing.

(Response) Thanks. Added in Figure 4.

Discussion: please add your responses to my previous questions in the discussion section to improve it and support your data.

(Response) Thanks for your comments. Added in Discussion.

Minor revisions:

Fig. 1C: X axis is not correct, please set a thick interval and adjust the graph accordingly.

(Response) Thanks for your kind comments. Corrected as column bar graph.

Round 3

Reviewer 1 Report

The concept of study design is logic and the results are reasonably to make the conclusion

Reviewer 3 Report

The manuscipt by Cho et al. has been significantly improved and can now be accepted for publication in IJMS.